# Multi-Omics Analysis of *SOX4*, *SOX11*, and *SOX12* Expression and the Associated Pathways in Human Cancers

**DOI:** 10.3390/jpm11080823

**Published:** 2021-08-23

**Authors:** Jaekwon Seok, Minchan Gil, Ahmed Abdal Dayem, Subbroto Kumar Saha, Ssang-Goo Cho

**Affiliations:** Department of Stem Cell and Regenerative Biotechnology, Incurable Disease Animal Model & Stem Cell Institute (IDASI), Konkuk University, 120 Neungdong-ro, Gwangjin-gu, Seoul 05029, Korea; tjrwornjs@naver.com (J.S.); minchangil@gmail.com (M.G.); ahmed_morsy86@yahoo.com (A.A.D.); subbroto@konkuk.ac.kr (S.K.S.)

**Keywords:** *SOX4*, *SOX11*, *SOX12*, prognosis, multi-omic analysis, oncogene

## Abstract

The Sry-related HMG BOX (SOX) gene family encodes transcription factors containing highly conserved high-mobility group domains that bind to the minor groove in DNA. Although some SOX genes are known to be associated with tumorigenesis and cancer progression, their expression and prognostic value have not been systematically studied. We performed multi-omic analysis to investigate the expression of SOX genes in human cancers. Expression and phylogenetic tree analyses of the SOX gene family revealed that the expression of three closely related SOX members, *SOX4*, *SOX11*, and *SOX12*, was increased in multiple cancers. Expression, mutation, and alteration of the three SOX members were evaluated using the Oncomine and cBioPortal databases, and the correlation between these genes and clinical outcomes in various cancers was examined using the Kaplan–Meier, PrognoScan, and R2 database analyses. The genes commonly correlated with the three SOX members were categorized in key pathways related to the cell cycle, mitosis, immune system, and cancer progression in liver cancer and sarcoma. Additionally, functional protein partners with three SOX proteins and their probable signaling pathways were explored using the STRING database. This study suggests the prognostic value of the expression of three SOX genes and their associated pathways in various human cancers.

## 1. Introduction

Cancer is one of the leading causes of death worldwide and a major threat to human health. Approximately 24.5 million new cancer cases and 9.6 million cancer deaths were reported in 2017 [1]. Although there has been substantial improvement in diagnostic and treatment modalities, the cumulative risk of cancer death between birth and 74 years of age remained at approximately 10% in 2018 [2]. Oncogenic processes include accumulated mutations and changes in gene expression, which result in gain of tumor traits such as limitless proliferation, invasion, metastasis, and immortalization [3]. Therefore, identification of differentially expressed genes (DEGs) between cancers and their normal counterparts, which are related to patient survival, can be exploited as therapeutic targets and diagnostic or prognostic markers for cancer.

The Sry-related HMG BOX (SOX) gene family consists of 20 transcription factors, which have a highly conserved high-mobility group (HMG) domain that binds to the minor groove in DNA [4]. During embryonic development, the SOX family members play essential roles as regulators of specific lineage and tissue gene expression for stemness, cell differentiation, organogenesis, and sex determination [5]. Some SOX members play oncogenic roles in various cancers. Upregulation of *SOX2* has been reported to be associated with poor clinical outcomes in breast [6], lung [7], renal [8], ovarian [9], and liver [10] cancers. Additionally, high expression of *SOX4* is associated with poor prognosis in leukemia, bladder [11], and breast [12,13] cancers [14]. Highly upregulated expression of *SOX9* is correlated with cancer progression in renal [15], liver [16], and colorectal [17] cancers. Furthermore, the relative mRNA expression levels of other SOX family members, such as *SOX11* and *SOX12*, are elevated in breast [18,19,20], liver [21,22], and lung [23,24] cancers and have a positive correlation with poor prognosis. In contrast, *SOX15* is a prospective tumor suppressor gene associated with the Wnt/β-catenin pathway in pancreatic cancer [25].

In this study, we systemically analyzed the expression of SOX family genes and their prognostic value in various cancers, using online bioinformatic databases. The expression of genes related to *SOX4*, *SOX11*, and *SOX12*, which are highly upregulated in various types of cancers, and their related signaling pathways were also examined using a gene meta-analysis database. The findings from these systemic analyses suggest the prognostic value of the expression of several SOX genes in human cancers and their potential as biomarkers for SOX gene-targeted cancer therapy.

## 2. Results

### 2.1. mRNA Expression of the SOX Gene Family in Various Cancers

The mRNA expression of the SOX gene family in different cancers and the corresponding normal tissues was analyzed using the Oncomine database (Version 4.5, Thermo Fisher, Waltham, MA, USA) with the following threshold parameters: *p*-value < 0.001, fold change of 2, and gene rank of 10%. Among the genes in the SOX family, the expression of *SOX4*, *SOX9*, *SOX11*, and *SOX12* was higher in multiple types of cancers than in their normal counterparts. (Figure 1a).

The amino acid sequences of SOX family proteins were retrieved from the National Center for Biotechnology Information (NCBI) database and clustered to create a phylogenetic tree, employing the neighbor-joining method using the Molecular Evolutionary Genetics Analysis (MEGA) tool (Figure 1b). Phylogenetic tree analysis showed that the highly expressed SOX members in various cancers, i.e., *SOX4*, *SOX11*, and *SOX12*, were closely clustered, suggesting similarities in amino acid sequences among these three SOX members. Although expression of *SOX9* was higher in various cancer types than their counterparts from the Oncomine database, the homology of *SOX9* was not strictly grouped with other three SOX members. Accordingly, we selected the three SOX genes *SOX4*, *SOX11*, and *SOX12* for further systematic analysis to determine their prognostic value and possible oncogenic role, using publicly available online databases.

### 2.2. SOX4 Expression and Its Prognostic Value in Various Cancers

In the Oncomine database, *SOX4* expression was upregulated in most types of cancer except melanoma and ovarian cancer (Figure 1a). In addition, the relative mRNA expression of *SOX4* in breast, liver, ovarian, pancreatic, and thyroid cancer as well as sarcoma datasets in the Oncomine database, was visualized (Figure 2a). We also found the expression of *SOX4* were significantly upregulated regardless of the stage of cancer including COAD, LIHC, LUAD, and LUSC (Appendix A). To investigate the protein expression level of *SOX4*, we accessed the CPTAC dataset using the UALCAN database (Preston, Lancashire, UK). In the CPTAC dataset, protein expression of *SOX4* was highly increased in breast cancer, uterine corpus cancer, and lung cancer (Appendix A). These results indicated that the mRNA and protein expression levels of *SOX4* were higher in three types of cancer tissues than in these neighboring normal tissues. Next, the mutation and copy number alterations of *SOX4* were examined in various cancer types using the cBioPortal database (version 3.7.2, MSKCC, New York, NY, USA) (Figure 2b). The locations of mutations in *SOX4* were distributed widely across the coding region. The alteration frequency of *SOX4* was determined in various types of cancers. The results showed that *SOX4* mutations were present in several cancer types, particularly in prostate and bladder cancer, with alteration frequencies of approximately 2% and 1.2%, respectively. Additionally, the alterations in *SOX4*, such as mutation, amplification, and deep deletion, showed a high percentage in various cancer types, including bladder, prostate, and ovarian cancer datasets, with alteration frequencies of 17%, 16%, and 14%, respectively. Further, the association between mRNA expression and clinical outcome was analyzed to determine the prognostic value of *SOX4* expression in patients with various cancers, using the online survival analysis tools such as the Kaplan–Meier plotter (Balazs Gyorffy, Budapest, Hungary), PrognoScan (Kyushu Institute Technology, Kyushu, Japan), and R2 databases (the Academic Medical Center (AMC), Amsterdam, the Netherlands). In these databases, a positive correlation between *SOX4* expression and poor patient survival was observed in colon, liver, lung, and pancreatic cancers, as well as sarcoma (Figure 2c). Overall, these results showed that the expression of *SOX4* was increased and correlated with poor patient survival in various cancers, including colon, liver, lung, and pancreatic cancers and sarcoma. 

### 2.3. SOX11 Expression Pattern and Patient Survival in Various Cancers

It is well known that *SOX11* expression may affect the progression of several types of cancers. Previously, elevated *SOX11* expression was observed in various cancer types, including brain, breast, head and neck, and gastric cancers [18,19,26,27]. However, the correlation between *SOX11* expression and patient survival has not yet been systematically investigated. The mRNA expression of *SOX11* was significantly higher in brain, breast, kidney, lung, and pancreatic cancers as well as sarcoma (Figure 3a). In addition, *SOX11* expression was significantly higher according to the stage of cancer, including BRCA, KIRC, and LUAD, than adjacent normal tissue from TCGA database using the UALCAN (Appendix A). Mutations in *SOX11* frequently occurred in esophageal, stomach, and lung cancers with D233 del/2_D233 del hotspot. In addition, the highest alteration frequency (14%) was observed in neuroendocrine prostate cancer (NEPC), wherein amplification was the major form of alteration (Figure 3b). Analysis of the correlation of survival and *SOX11* expression using Kaplan–Meier plotter, PrognoScan, and R2 databases revealed significantly higher survival in the low *SOX11* expression group of cancer patients than in the high expression group (Figure 3c). These results suggest that upregulation of *SOX11* is correlated with poor prognosis in various types of cancer.

### 2.4. SOX12 Expression Pattern and Patient Survival in Various Cancers

*SOX12* has been actively studied in recent years and is known to affect cancer characteristics. For example, the upregulation of *SOX12* has been reported to promote cancer proliferation or migration in colorectal, lung, liver, gastric, and breast cancers [20,22,24,28,29]. Increased *SOX12* expression was observed in breast, esophageal, lung, and ovarian cancers as well as sarcoma, compared to that in their corresponding normal tissues, using the Oncomine database (Figure 4a). In addition, the mRNA expression of *SOX12* was significantly upregulated in BRCA, ESCA, LUSC, and LUAD regardless of the cancer stage compared to their normal tissues (Appendix A). *SOX12* mutation mainly occurred in bladder cancer with a hot spot of E294K, but there were a few comparable differences with the other cancer types. Moreover, alterations in *SOX12* were the highest in neuroendocrine prostate cancer, with an alteration frequency of 20% (Figure 4b). Survival curve analysis, using the PrognoScan, Kaplan–Meier plotter, and R2 databases, showed that the high-expression group had a significantly lower survival rate than the low-expression group (Figure 4c). Overall, these data suggest that the regulation of *SOX12* expression in several types of cancers is significantly related to patient survival.

### 2.5. Clinical Prognosis of the Three SOX Genes in Liver Cancer

The analysis of *SOX4*, *SOX11*, and *SOX12* expression revealed that their expression was correlated with patient survival in several types of cancers. To investigate the correlation between clinical prognosis and the co-expression of *SOX4*, *SOX11*, and *SOX12*, we used the Kaplan–Meier plotter database, wherein the overall survival of patients was compared between the two groups, classified based on the average expression level of the three SOX genes. Survival plot analysis of the three SOX genes was performed using pan-cancer RNA sequencing datasets in the Kaplan–Meier plotter. In this study, we found that co-expression of the three SOX genes in liver cancer was significantly related to poor overall survival of patients, indicating negative correlation of each SOX gene with overall survival (Figure 5a). Next, to investigate the correlation between expression of the three SOX genes in liver cancer, we examined their transcriptomic datasets in TCGA database using the cBioPortal platform. Correlation heatmap analysis was performed using the Pearson score of the three SOX genes (Figure 5b). The three SOX genes showed positive correlation with each other in the liver hepatocellular carcinoma dataset (TCGA-LIHC). To examine commonly related pathways of the three SOX genes, positively and negatively co-altered genes of each SOX gene in TCGA-LIHC dataset were retrieved and 404 commonly positively co-altered genes (Figure 5c) and 318 negatively co-altered genes (Appendix A) between the three SOX genes were represented by a Venn diagram. The Reactome pathway analysis with positively commonly correlated genes among the three SOX genes revealed pathways related to the cell cycle and Golgi to ER transport (Figure 5d). In the pathway analysis with commonly downregulated genes, certain correlated genes were classified in metabolic pathways, including metabolism of lipids, amino acids, and other macromolecules (Appendix A). Overall, these results suggest that the three SOX genes are commonly correlated with patient survival and are associated with several key pathways involved in cancer progression.

### 2.6. Clinical Prognosis of the Three SOX Genes in Sarcoma

Furthermore, we examined the prognostic value of co-expression of the three SOX genes in sarcoma. The overall survival of patients was significantly correlated with the co-expression of the three SOX genes, as well as the expression of individual SOX genes (Figure 6a). Survival analysis with co-expression of SOX genes indicated a lower *p*-value (0.00014) than the *p*-values (0.0029, 0.01, and 0.015, respectively) of individual SOX genes, thereby suggesting a stronger prognostic value of co-expression of the three SOX genes than individual SOX expression. Subsequently, to investigate the correlation of all SOXC members in sarcoma, we examined the transcriptome datasets of sarcomas using TCGA database through the cBioPortal platform. In the heatmap analysis using Pearson score, we found that the three SOX genes were positively correlated with each other in the sarcoma dataset (TCGA-SARC) (Figure 6b). Next, we aimed to identify commonly related pathways involving the three SOX genes that might play an oncogenic role in sarcoma. To investigate the genes correlated with the three SOX genes, we found a co-altered gene set with each SOX gene from the TCGA-SARC dataset through the R2 platform, and commonly co-altered genes were represented by a Venn diagram (Figure 6c). Compared to those in liver cancer, only 16 genes were commonly upregulated with the three SOX genes in sarcoma, as indicated in the Venn diagram, while 9 genes were commonly negatively correlated (Appendix A). The total commonly correlated DEGs were classified using the Reactome pathway analysis (Version 76, Ontario Institute for Cancer Research, Toronto, ON, Canada), which revealed pathways related to DNA methylation, exostosis, ephrin signaling, glycosaminoglycan metabolism, and telomere extension (Figure 6d). In addition, the Reactome pathway analysis with commonly downregulated genes revealed ontology terms such as neurodegenerative disease, FOX-mediated transcription, and immune system, including interleukin-12 signaling (Appendix A). Taken together, expression of the three SOX genes could have prognostic value and might be associated with certain essential pathways related to the immune system, defective glucuronosyltransferase activity, and DNA methylation, which result sarcoma progression.

### 2.7. Functional Protein Partners and Their Predicted Signaling Pathways

To investigate the three SOX gene-related pathways that might play a role in various types of cancers, 40 proteins that commonly correlated with the three SOX genes were retrieved from the STRING database (version 11, Swiss Institute of Bioinformatics, Lausanne, Switzerland) (Figure 7). The predicted interacting proteins that showed high confidence included TP53, POU3F2, POU3F3, TCF7, TCF7L1, TCF7L2, CTNNB1, and PTEN. Next, association of the 40 proteins was analyzed to predict probable signaling pathways and gene ontology (GO) categories. The KEGG pathway analysis revealed that individual functional protein partners were categorized in pathways related to the Hippo signaling pathway, WNT signaling pathway, cell cycle signaling pathway, p53 signaling pathway, and several other cancer-related pathways (Table 1). In addition, the Reactome pathway analysis showed that some categories were related to TCF, β-catenin, and the WNT signaling pathway (Table 2). The GO categories obtained from the analysis using extracted functional protein partners, also contained terms related to the WNT signaling pathway, TCF/β-catenin complex, and other developmental processes. Furthermore, gene network analysis of three SOX genes was determined from the GeneMANIA database. In the GeneMANIA webtool, 20 genes showed high confidence with the SOXC members included SMARCA4, WT1, NF1, KRAS, GATA3, TERT, POU3F2, POU3F3, SOX3, and SOX5 (Appendix A). These findings suggest that the three SOX genes could be associated with specific critical pathways related to certain developmental processes, including the WNT signaling pathway and the TCF/βcatenin complex, in cancer progression.

Functional protein partners of *SOX4*, *SOX11*, and *SOX12* were predicted using the Cytoscape string application. The line color implies that each SOX protein interacts separately with other proteins. The node and line colors of each SOX gene were the same. *SOX4*: red color; *SOX11*: green color; *SOX12*: navy color.

## 3. Discussion

Members of the SOX gene family are essential transcription factors for human developmental processes such as sex determination, cell differentiation, and organogenesis [3]. Nevertheless, it has been reported that the SOX family may participate in tumor progression as transcriptional activators or repressors depending on their specific binding partners [30]. In the present study, we revealed that phylogenetically close SOX members, *SOX4*, *SOX11*, and *SOX12*, have distinctively higher expression in various cancers, and analyzed the prognostic value and correlated pathways using various bioinformatic tools. These three SOX proteins belong to the SOXC subgroup and are evolutionarily conserved in vertebrates [30,31]. Although it has been reported previously that higher expression of *SOX4*, *SOX11*, and *SOX12* is negatively correlated with patient survival in various cancers [9,10,11,12,13,14,15,16,17,18,19,20,21,22], the expression of SOXC members has not yet been analyzed systematically in multiple cancer datasets. In most types of cancer, SOXC members could serve as the oncogenic role and poor prognostic value, however previous studies have been reported that the function of *SOX11* was associated both with an oncogenic or tumor-suppressive role in carcinoma [26,32]. In our study, the higher expression of three SOX genes was determined in various types of cancer regardless of their cancer stage compared to normal tissues. Our systematic analysis demonstrated that the expression of the three SOXC members was distinctively higher in various types of cancers than in their normal counterparts, thereby indicating a negative correlation of these proteins with patient survival in multiple cancer datasets. In particular, the poor prognostic value of *SOX11* expression in pancreatic cancer and sarcoma and *SOX12* expression in breast and ovarian cancer as well as sarcoma, has not been reported previously (Figure 2, Figure 3 and Figure 4). We also found strong correlations between mRNA and *SOX4* protein expression in CPTAC datasets including breast, uterine corpus, and lung cancer from TCGA database. However, Protein expression of *SOX11* and *SOX12* was not detectable or significant in CPTAC datasets. In the Oncomine database, *SOX9* was also highly expressed in multiple types of cancers than healthy counterparts and it has been reported previously that higher expression of *SOX9* is correlated with poor prognosis in several cancers [15,16,17]. Although *SOX9* has strong prognostic value in various types of cancers, phylogenetic tree analysis of the SOX gene family proteins has shown weak homology of *SOX9* with SOXC members. Correlation between expression of *SOX9* and poor patient prognosis strongly suggests that *SOX9* might have an oncogenic role in several types of cancer, which remains to be explained in further studies.

Co-expression of all SOXC members largely occurs in mouse embryos, mid-organogenesis, and several tissues [31,33,34,35]. We also observed co-expression of SOXC members in the LIHC and SARC datasets (Figure 5b and Figure 6b). The increased expression of these three SOX members in cancers and sequence similarities raise the question that whether the expression of the three SOX genes has possible compensatory or synergic role in cancer progression. We examined the effect of co-expression of *SOX4*, *SOX11*, and *SOX12* on patient survival in TCGA datasets. In TCGA-LIHC and TCGA-SARC datasets, co-expression of the three SOX genes was positively correlated with poor outcomes. We also observed that the expression levels of the three SOX members were positively correlated with each other in the LIHC and SARC datasets. In the SARC dataset, the *p*-value for correlation between co-expression of the three SOX members and patient survival was much less than that for correlation with individual SOX members, indicating that co-expression of the three SOX members has a stronger prognostic value than the expression of individual SOX members. The stronger prognostic value of co-expression than with each SOX member might reflect the synergic effect of the three SOX members. However, in the LIHC dataset, co-expression did not indicate a significantly higher prognostic value for patient survival. This indicates that co-expression of SOX proteins is more predictive in SARC than in LIHC.

The consistent prognostic value associated with the expression of the three SOX genes in cancers suggests shared functional pathways between these SOX genes in various cancer types. We analyzed the shared pathways among the genes co-expressed with *SOX4*, *SOX11*, and *SOX12* in the LIHC and SARC datasets. In LIHC, the 404 commonly co-expressed genes (9.5% of analyzed correlated genes) were highly correlated with the cell cycle and mitosis-related pathways. However, in SARC, only 16 genes (1.4% of analyzed correlated genes) were identified as commonly significantly correlated genes, which were mainly related to pathways involved in epigenetic regulation. These commonly related pathways were highly associated with cancer progression. Lower number of commonly-related genes among the three SOX genes in SARC may reflect that independent pathways of each SOX member could cooperatively work, thereby leading to cancer progression in SARC, which could explain the stronger prognostic value of the three SOX genes than that of individual SOX genes in SARC.

PPI network analysis of the three SOX proteins revealed their functional protein partners to be categorized in several cancer-related pathways, including the TNF/β-catenin and WNT signaling pathways. Previous studies have described a single SOX gene-related canonical WNT/β-catenin signaling pathway in various cancers [36,37]. However, our analysis is the first to reveal a correlation between the three SOX genes and a group of genes that are related to several pathways. Our findings indicate an association between expression of the three SOX genes and their predicted signaling pathways in some cancers. In addition, our results of the PPI network analysis showed similar signaling pathways with correlated genes, including the cell cycle and SUMOylation, in LIHC and SARC. We also analyzed the gene network analysis of *SOX4*, *SOX11*, and *SOX12* using GeneMANIA database. In this result, these commonly related genes including TERT, KRAS, and CDKN2A were also highly associated with cancer progression. Previous studies have reported that regulation of TERT, KRAS, and CDKN2A could serve as potential therapeutic targets in multiple cancer types [38,39,40]. However, the mechanisms of action of these genes in cancer progression remain unknown and need to be elucidated in the future.

The alterations in the three SOX genes were investigated to determine which types of cancers were related to significant alterations in the SOX members, using cBioPortal. Mutations in oncogenes, such as TP53, KRAS, and PIK3AC, are correlated with clinical outcomes in various cancers [41,42,43,44]. Alterations are associated with human cancers, and regions of structural variation in the human genome can be novel biomarkers for cancer progression [45]. In our study, we found amplification of *SOX4* and their upregulated expression in multiple cancer types. In addition, previous studies have reported that amplification of *SOX4* is associated with cancer progression [46,47,48]. Although the amplification of *SOX4* can lead to upregulation of *SOX4* expression, the association between mutations of *SOX4* and their gene expression with various cancer phenotypes was unknown and should be pursued in further study. Indeed, the *SOX11* and *SOX12* gene-altered patient group had poorer survival than the unaltered group in the SARC dataset (unpublished data). However, the functional importance of mutations and alterations in *SOX4*, *SOX11*, and *SOX12* remain unknown. In addition, more experimental and theoretical studies are recommended to support the outcomes of this study, since the clinical data mining-based analyses need to validate the underlying molecular mechanisms.

## 4. Materials and Methods

### 4.1. Oncomine Database Analysis

The differential mRNA expression of the SOX family members in various cancer tissues versus their normal counterparts, was examined using the Oncomine database (Thermo Fisher, Waltham, MA, USA) (https://www.oncomine.org/ (accessed on 11 May 2021); version 4.5) [49,50]. Fold-change in the mRNA expression of the SOX family genes in cancer tissues, compared to that in their normal counterparts, was calculated based on a threshold *p*-value < 0.05; fold-change of 2.

### 4.2. Molecular Evolutionary Genetics Analysis

The amino acid sequences of the SOX family proteins were retrieved from the NCBI database. Accession numbers of the SOX protein sequences are mentioned in Appendix A. Multiple sequence alignment was performed using the ClustalW program from the Molecular Evolutionary Genetics Analysis X (MEGA-X) tool. Multiple-aligned sequences were used to build a phylogenetic tree by employing the neighbor-joining method.

### 4.3. GEPIA2 Database Analysis

Gene expression profiling interactive analysis 2 (GEPIA2) (Peking University, Beijing, China) (http://gepia2.cancer-pku.cn/ (accessed on 11 May 2021)) is an online bioinformatic tool for analyzing RNA expression using The Cancer Genome Atlas (TCGA) data [51]. In this study, GEPIA2 was used to analyze the expression of *SOX4*, *SOX11*, and *SOX12* and their association with the survival of patients in multiple cancer types. Differential gene expression between TCGA tumor samples and a combination of TCGA normal samples and Genotype-Tissue Expression (GTEx) normal samples, was visualized using boxplots. A survival curve was generated, using GEPIA, to determine the association of SOX gene expression with patient survival in different cancer types. The correlation between SOX gene expression and patient survival was analyzed using Kaplan–Meier survival curves and log-rank test using GEPIA2.

### 4.4. cBioPortal Database Analysis

cBioPortal (MSKCC, New York, NY, U.S.A.) (http://www.cbioportal.org (accessed on 11 May 2021); version 3.7.2) is a web-based genomic portal that provides visualization and analysis of TCGA datasets [52,53]. In this study, cBioPortal was used to analyze the mutation and alteration frequency of SOX genes with relevant parameter settings.

### 4.5. Kaplan–Meier Plotter

The Kaplan–Meier plotter (Balazs Gyorffy, Budapest, Hungary) (http://kmplot.com/analysis/ (accessed on 11 May 2021)) is a web-based database that provides survival curve of patients, based on 54,675 genes in 21 cancer types [54]. Comparison of survival between the two patient groups, classified according to the expression level of each SOX gene, with “auto select best cutoff” option was carried out using the survival curve. The survival curve indicating the co-occurrence of SOX genes was retrieved to investigate the prognostic value of co-expression of SOX genes in pan-cancer RNA sequencing data. Survival analysis was performed to determine the average expression levels of *SOX4*, *SOX11*, and *SOX12*.

### 4.6. R2: Genomic Analysis and Visualization Platform

The R2 platform (the Academic Medical Center (AMC), Amsterdam, Netherlands) (https://hgserver1.amc.nl/cgi-bin/r2/main.cgi (accessed on 11 May 2021)) is a publicly available web-based genomic analysis and visualization platform that uses information from the TCGA, Gene Expression Omnibus (GEO), and GTEx projects. In this study, we performed survival analysis of the mRNA expression of the three SOX genes in several types of cancers, using the R2 online tools.

### 4.7. Gene Correlation Analysis of SOX4, SOX11, and SOX12

Correlated genes of the three SOX members were determined from TCGA datasets of liver cancer and sarcoma, using the R2 platform (https://hgserver1.amc.nl/cgi-bin/r2/main.cgi (accessed on 11 May 2021)). The analysis was performed with the adjustment of the Bonferroni test using a threshold *p*-value < 0.01.

Next, Venn diagrams were used to identify common genes among the correlated genes of *SOX4*, *SOX11*, and *SOX12* in liver cancer and sarcoma, using Venny 2.1.0 (Spanish National Biotechnology Centre (CNB)-CSIC, Madrid, Spain) (https://bioinfogp.cnb.csic.es/tools/venny/ (accessed on 11 May 2021)).

To explore pathways associated with the genes commonly correlated with *SOX4*, *SOX11*, and *SOX12*, we used the Reactome pathway database (Ontario Institute for Cancer Research, ON, USA) (https://reactome.org/ accessed on 11 May 2021; version 76) [55].

### 4.8. Identification of Functional Protein Partners of *SOX4*, *SOX11*, and *SOX12*, and Signaling Pathway Analysis

The functional protein partners of *SOX4*, *SOX11*, and *SOX12* were analyzed using STRING database v11.0 (https://string-db.org/ (accessed on 11 May 2021)). Subsequently, these data were reorganized using the Cytoscape tool to visualize the interaction network of each protein. Pathway and gene ontology (GO) analyses were performed using the selected protein partners. Pathway analysis was classified based on the KEGG and REACTOME pathway databases [56].

### 4.9. Analysis of SOX Protein Expression Pattern in Various Types of Cancer

The protein expression levels of SOX protein in multiple cancer types were investigated from the UALCAN databases (Preston, Lancashire, UK) (https://ualcan.path.uab.edu/index.html (accessed on 11 May 2021)) [57]. Protein expression level of *SOX4* was systematically analyzed according to cancer stage on the characteristics of patients with COAD, LIHC, LUAD, and LUSC, derived from the Clinical Proteomic Tumor Analysis Consortium (CPTAC). Differences with *p*-value < 0.05 were considered statistically significant.

### 4.10. Analysis of Gene Network with SOX4, SOX11, and SOX12

The gene network analysis with *SOX4*, *SOX11*, and *SOX12* was conducted using GeneMANIA database (University of Toronto, TN, Canada) (https://genemania.org/ (accessed on 11 May 2021)) [58]. In this study, GeneMANIA was used to analyze the interaction gene network with three SOX genes.

## 5. Conclusions

In this multi-omics analysis of SOX genes expression in human cancer databases, we suggest the evidence of the correlation between the expression of three SOX genes and clinical outcomes in human cancer. Our study provides the importance of all SOXC members expression and possible three SOX genes related pathways in cancer progression. Therefore, our analysis may contribute valuable insights into *SOX4*, *SOX11*, and *SOX12* as a potential therapeutic goal for various human cancers.

## Figures and Tables

**Figure 1 jpm-11-00823-f001:**
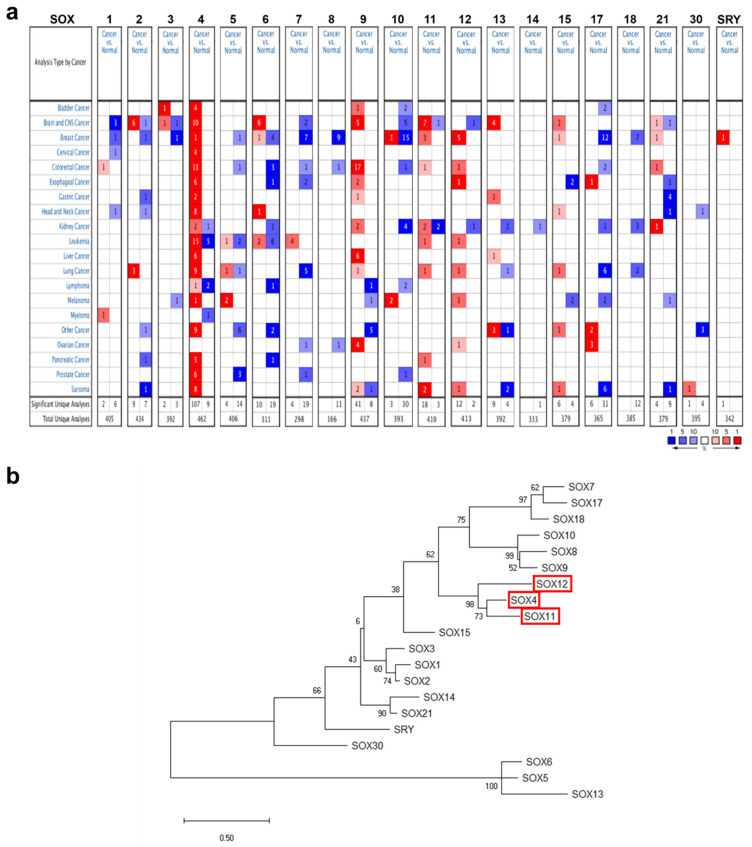
mRNA expression levels of the SOX gene family in different cancer types. (**a**) Comparison of the expression of SOX gene family members from the Oncomine database, indicating the number of datasets with mRNA overexpression (red) or underexpression (blue), between various types of cancers and their normal counterparts. The threshold was determined based on the following parameters: *p*-value < 0.001, fold-change of 2, and gene ranking of 10%. (**b**) Phylogenetic tree analysis of the SOX gene family proteins was performed using the Molecular Evolutionary Genetic Analysis (MEGA) tool. The tree was derived from the maximum likelihood method.

**Figure 2 jpm-11-00823-f002:**
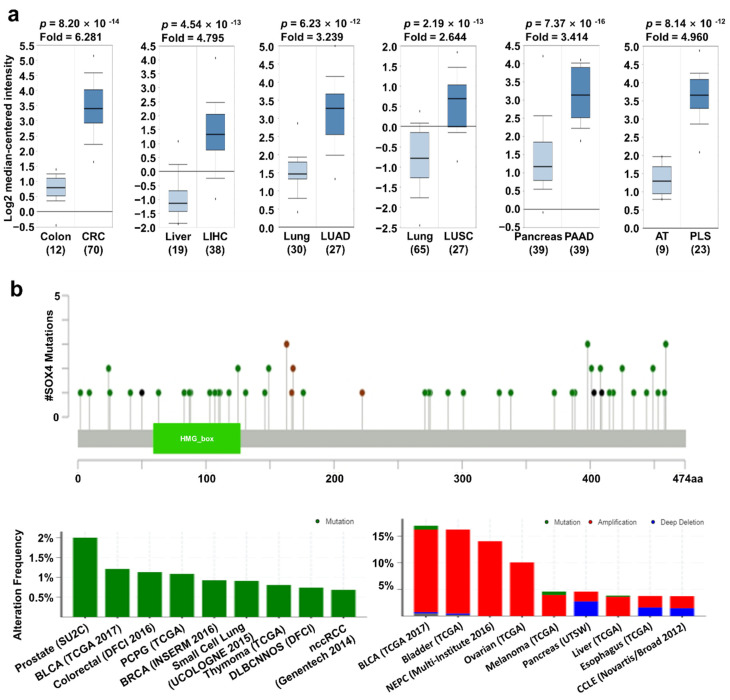
*SOX4* expression pattern and patient survival in various cancers. *SOX4* expression, alteration frequency of mutations, copy number alteration, and correlation with patient survival in various types of cancer. (**a**) Box plots comparing SOX4 expression in normal (left plot) and cancer tissues (right plot) were derived from the Oncomine database. The fold-changes in *SOX4* expression in colon, lung, liver, and pancreatic cancers, as well as sarcoma are shown as box plots. (**b**) The mutations and alteration frequency of *SOX4* were determined using the cBioPortal database. The diagram shows *SOX4* mutation in different cancer types across protein domains. The alteration frequency of *SOX4* was retrieved using cBioPortal and is shown at the bottom. Only those cancer types with more than 100 samples and an alteration frequency of over 3.7% are shown. The alterations comprised deletions (blue), amplification (red), or mutation (green). (**c**) The survival plot comparing patients with high (red) and low (blue) expression in various cancer types was plotted using the Kaplan–Meier plotter, PrognoScan, and R2 databases. Survival plot analysis was performed using a threshold Cox *p*-value of <0.05. Abbreviations. CRC: colorectal cancer; LIHC: liver hepatocellular carcinoma; LUAD: lung adenocarcinoma; LUSC: lung squamous cell carcinoma; PAAD: pancreatic adenocarcinoma; AT: adipose tissue; PLS: pleomorphic liposarcoma.

**Figure 3 jpm-11-00823-f003:**
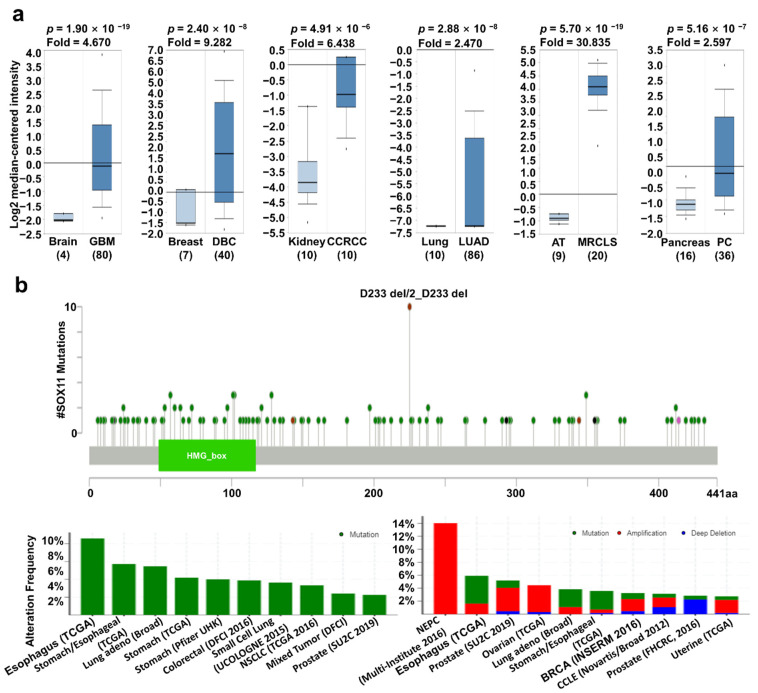
*SOX11* expression pattern and patient survival in various cancers. *SOX11* expression pattern, alteration frequency of mutations, copy number alteration, and patient survival in various types of cancers. (**a**) Box plots comparing *SOX11* expression in normal (left plot) and cancer tissues (right plot) were derived from the Oncomine database. The fold-changes in *SOX11* expression in brain, breast, lung, kidney, and pancreatic cancers as well as sarcoma, are shown as box plots. (**b**) Mutations and alteration frequency of *SOX11* were determined using the cBioPortal database. The diagram shows *SOX11* mutations in different cancer types across protein domains. *SOX11* mutations mainly occurred in esophageal cancer with one hot spot (D233 del/2_D233 del) representing common founder mutations. The alteration frequency of *SOX11* was determined using cBioPortal and is shown at the bottom. Only those cancer types with more than 100 samples and an alteration frequency of >2.73% are shown. The alterations comprised deletions (blue), amplification (red), or mutation (green). (**c**) Survival plots comparing patients with high (red) and low (blue) *SOX11* expression in various cancer types were plotted using the Kaplan–Meier plotter, PrognoScan, and R2 databases. Patient survival analysis was performed using a threshold Cox *p*-value of <0.05. Abbreviations. GBM: glioblastoma; DBC: ductal breast carcinoma; CCRCC: clear cell renal cell carcinoma; MRCLS: myxoid/round cell liposarcoma; PC: pancreatic carcinoma.

**Figure 4 jpm-11-00823-f004:**
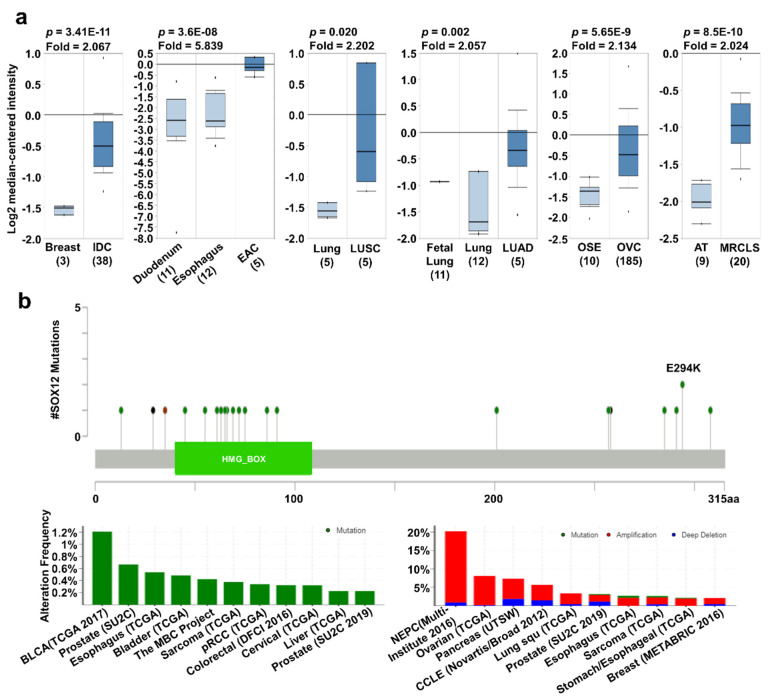
*SOX12* expression pattern and patient survival in various cancers. *SOX12* expression pattern, alteration frequency of mutations, copy number alterations, and patient survival in various types of cancers. (**a**) Box plots comparing specific *SOX12* expression in normal (left plot) and cancer tissues (right plot) were derived from the Oncomine database. The fold-changes in SOX12 expression in breast, esophageal, lung, and ovarian cancers and sarcoma were determined and are shown as box plots. (**b**) The mutations and alteration frequency of *SOX12* were determined using the cBioPortal database. *SOX12* mutation diagram of different cancer types across protein domains is shown. *SOX11* mutation mainly occurred in bladder cancer with one hot spot (E294K) representing the common founder mutations. The alteration frequency of *SOX12* was determined using the cBioPortal database and is shown at the bottom. Only cancer types containing more than 100 samples and an alteration frequency of >2.1% are shown. The alterations comprised deletions (blue), amplification (red), or mutation (green). (**c**) The survival plot comparing patients with high (red) and low (blue) expression in various cancer types was obtained using the Kaplan–Meier plotter, PrognoScan, and R2 databases. Survival plot analysis was performed using a threshold Cox *p*-value of <0.05. Abbreviations. IDC: invasive ductal carcinoma; OSE: ovarian surface epithelium; OVC: ovarian carcinoma; EAC: esophageal adenocarcinoma.

**Figure 5 jpm-11-00823-f005:**
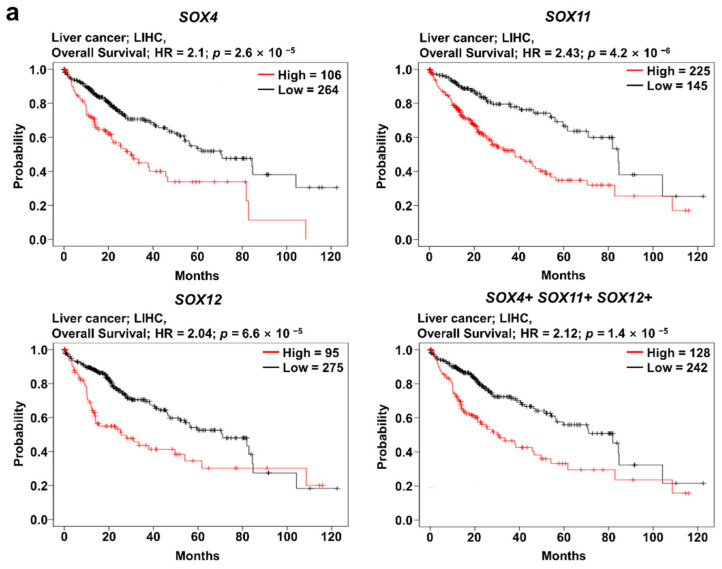
Prognostic value of co-expression of the three SOX genes in liver cancer. Correlation of *SOX4*, *SOX11*, and *SOX12* expression with the prognosis of liver cancer, and analysis of commonly positively correlated genes of the three SOX genes. (**a**) Survival curve comparing patients with high (red) and low (black) expression of each SOX gene using the Kaplan–Meier plotter. The plotters were analyzed for liver cancer. (**b**) Correlation heat map of *SOX4, SOX11,* and *SOX12* expression data, generated using TCGA-LIHC RNA sequencing data from the cBioPortal database. Pearson’s correlation was calculated among the three SOX genes to determine the co-expression pattern of genes in the heat map. (**c**) Venn diagram of the genes positively correlated with the three SOX genes, generated using the TCGA-LIHC transcriptome dataset from the R2 database. (**d**) Reactome pathway analysis of the genes positively correlated with the three SOX genes, using the TCGA-LIHC transcriptome dataset.

**Figure 6 jpm-11-00823-f006:**
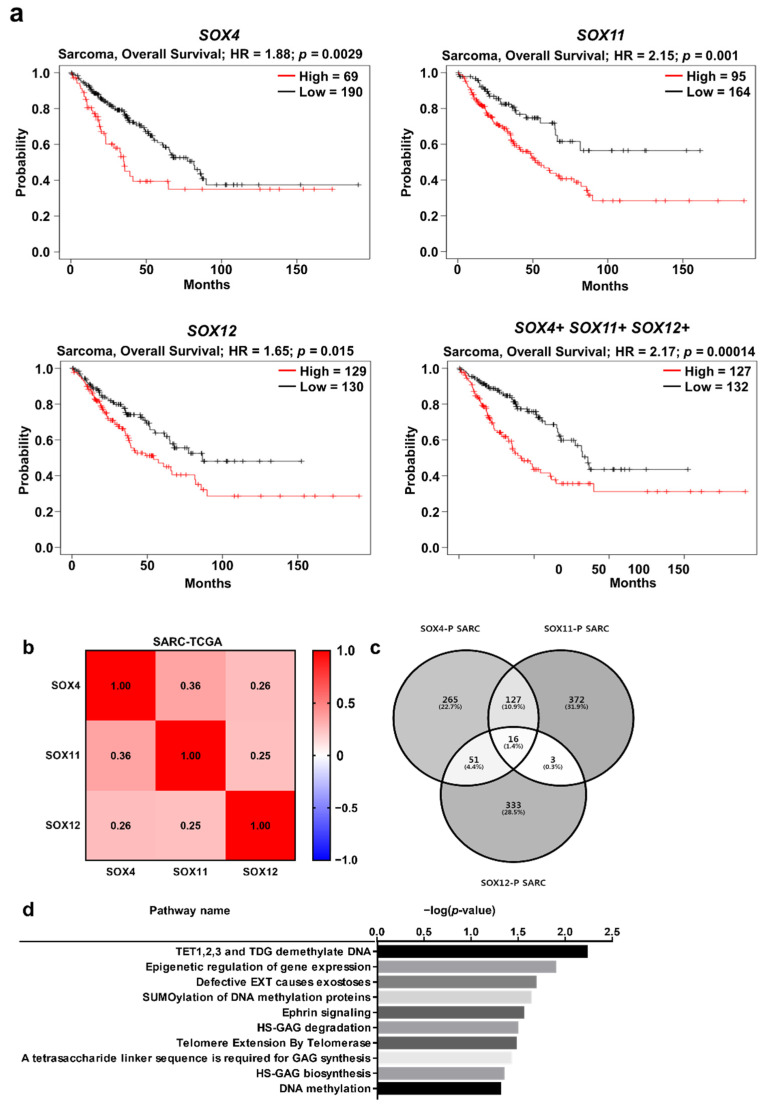
Correlation of *SOX4*, *SOX11*, and *SOX12* expression with the prognosis of sarcoma, and analysis of positively correlated genes with the three SOX genes. (**a**) Survival curve comparing patients with high (red) and low (black) expression of each SOX gene from the Kaplan–Meier plotter. The plotters were analyzed for sarcoma. (**b**) Correlation heat map of *SOX4, SOX11*, and *SOX12* expression data, generated using TCGA-SARC RNA sequencing data from cBioPortal database. Pearson’s correlation was calculated among the three SOX genes to determine the co-expression pattern of genes in the heat map. (**c**) Venn diagram of the genes positively correlated with the three SOX genes, generated using the -TCGA-SARC dataset from the R2 database. (**d**) Reactome pathway analysis of the genes positively correlated with the three SOX genes, using TCGA-SARC transcriptome dataset.

**Figure 7 jpm-11-00823-f007:**
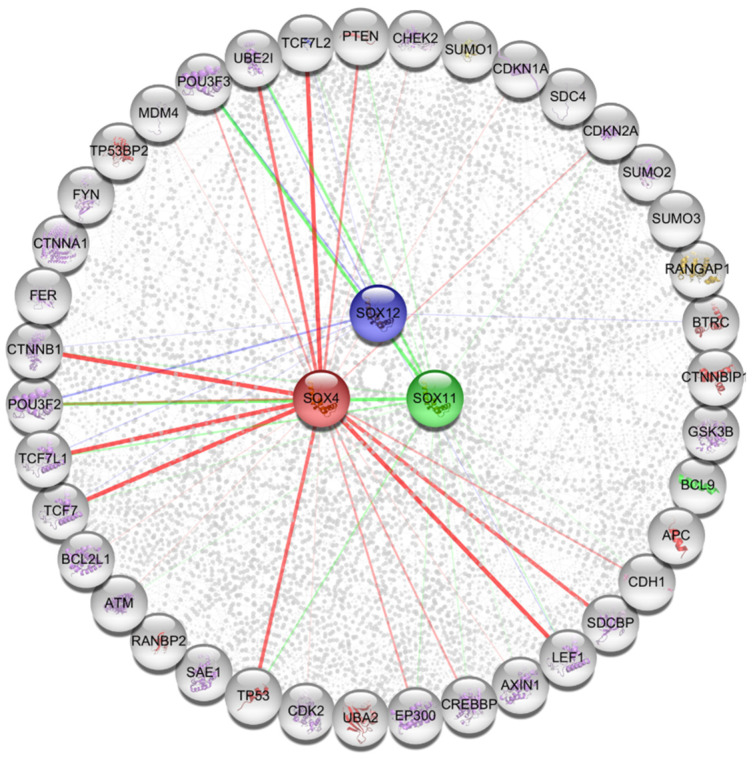
Functional protein partners and their predicted signaling pathways.

**Table 1 jpm-11-00823-t001:** KEGG pathway analysis of individual functional protein partners correlated with the three SOX genes.

#Term ID	Term Description	False Discovery Rate
hsa05213	Endometrial cancer	2.20 × 10^−20^
hsa05215	Prostate cancer	4.75 × 10^−18^
hsa05200	Pathways in cancer	2.24 × 10^−17^
hsa04310	Wnt signaling pathway	2.61 × 10^−16^
hsa05226	Gastric cancer	2.92 × 10^−16^
hsa04520	Adherens junction	3.34 × 10^−16^
hsa05225	Hepatocellular carcinoma	7.38 × 10^−16^
hsa05165	Human papillomavirus infection	2.76 × 10^−15^
hsa05217	Basal cell carcinoma	5.93 × 10^−15^
hsa05206	MicroRNAs in cancer	8.70 × 10^−15^
hsa04390	Hippo signaling pathway	9.90 × 10^−15^
hsa05210	Colorectal cancer	6.99 × 10^−14^
hsa05224	Breast cancer	2.64 × 10^−13^
hsa04934	Cushing’s syndrome	3.71 × 10^−13^
hsa05216	Thyroid cancer	4.14 × 10^−13^
hsa04115	p53 signaling pathway	4.94 × 10^−13^
hsa04110	Cell cycle	1.55 × 10^−12^
hsa05166	HTLV-I infection	4.71 × 10^−11^
hsa04916	Melanogenesis	4.15 × 10^−10^
hsa04218	Cellular senescence	4.27 × 10^−10^

**Table 2 jpm-11-00823-t002:** Reactome pathway analysis of individual functional protein partners correlated with the three SOX genes.

#Term ID	Term Description	False Discovery Rate
HSA-201681	TCF-dependent signaling in response to WNT	4.69 × 10^−15^
HSA-3769402	Deactivation of the β-catenin transactivating complex	4.69 × 10^−15^
HSA-8878159	Transcriptional regulation by RUNX3	1.63 × 10^−12^
HSA-212436	Generic Transcription Pathway	3.64 × 10^−12^
HSA-2990846	SUMOylation	5.69 × 10^−12^
HSA-74160	Gene expression (Transcription)	9.40 × 10^−12^
HSA-195253	Degradation of β-catenin by the destruction complex	1.42 × 10^−11^
HSA-162582	Signal Transduction	3.63 × 10^−11^
HSA-1640170	Cell Cycle	4.88 × 10^−11^
HSA-201722	Formation of the β-catenin/TCF transactivating complex	7.06 × 10^−11^
HSA-6804760	Regulation of TP53 activity through methylation	4.81 × 10^−10^
HSA-3065678	SUMO is transferred from E1 to E2 (UBE2I, UBC9)	6.14 × 10^−10^
HSA-4411364	Binding of TCF/LEF: CTNNB1 to target gene promoters	1.59 × 10^−9^
HSA-8951430	RUNX3 regulates WNT signaling	1.59 × 10^−9^
HSA-3108232	SUMO E3 ligases SUMOylate target proteins	1.86 × 10^−9^
HSA-69563	p53-dependent G1 DNA Damage Response	3.53 × 10^−9^
HSA-1643685	Disease	4.70 × 10^−9^
HSA-3700989	Transcriptional Regulation by TP53	6.39 × 10^−9^
HSA-5663202	Diseases of signal transduction	6.39 × 10^−9^
HSA-6804757	Regulation of TP53 Degradation	6.39 × 10^−9^

## Data Availability

The data presented in this study are available on request from the corresponding author.

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
