# Peer review of "Multi-Omics Analysis of SOX4, SOX11, and SOX12 Expression and the Associated Pathways in Human Cancers"

_jpm, 2021, doi:10.3390/jpm11080823_

Round 1

Reviewer 1 Report

The authors focused  systemically analyzedthe expression of SOX family genes and their prognostic valuein  various  cancers,using  online  bioinformatic  databases.  

I suggest to supplement the presented results with data analysis from the "Clinical Proteomic Tumor Analysis Consortium (CPTAC)" database. Conducting the analysis within the CPTAC-TCGA cohort, will allow to determine the correlations between mRNA and SOX proteins expression profile within the available tumors.

The analysis of the SOX9 gene would also be interesting, although during the analysis of the phylogenetic tree it was not strictly grouped with SOX4, SOX11, SOX12, however among the genes in the SOX family, the expression of  SOX9 washigher  in  multiple  types  of  cancers  than  in  their  normal counterparts.

The  survival  plot  comparing  patients  with  high   and  low    expression  in various  cancer types should be more readable (Figures:2c,3c,4c,5a,6a).

Author Response

Please see the attachment~

Reviewer 2 Report

The study by Jaekwon Seok and colleagues investigate, in a systematic manner, the expression of SOX genes in human cancers. Downstream analyses are focused on SOX4, SOX11 and SOX12 and unveiled their associated pathways and their relevance in predicting cancer survival in several cancers. These observations are based on the extraction of data from previously published datasets that are available on public servers (Oncomine, STRING, R2 and cBioportal). Authors used tools available on these platforms to retrieved gene expression data as well as produced Kaplan-Meier survival curve and identify the pathways associated with SOX genes. This information may be of interest to SOX aficionados and scientists involved in cancer research.

In its current status, the manuscript does not include: new bioinformatic or statistical method ; proper validation of the data on fresh cancer samples ; experimental data to further explore/validate some of the correlations. It is thus of limited interest as any scientist could access these databases on a regular basis to perform the same analysis. For instance, figures are often directly copy/paste from the interface of the databases.

It would thus be necessary to, at least, validate that some of the correlations and observations extracted from the public platforms are meaningful.

Reviewer 3 Report

Seok and colleagues present a bioinformatic based analysis of various genomics databases to assess the role of the SOX gene family in human cancers. The study is interesting and provides a potential map road to utilizing such publicly available databases to study the impact of other genes in cancer and other human disease phenotypes. My comments are listed below.  

  1. From Fig. 1, assuming I'm reading the heatmap correctly, it seems that among the rest of the SOX genes, SOX4 is the primary gene associated with cancer phenotype in different organs, with SOX11 and 12 showing a much weaker association compared to SOX4. Based on the heatmap, it would seem that the rest 19 SOX genes are not or at least very weakly associated with cancer. As the authors are well aware, cancer is a complex disorder characterized by multiple stages, each of which is associated with unique molecular processes (e.g., how SOX genes affect metastasis). To that end, have the authors looked at how the presence (e.g., expression) of the individual SOX genes change depending on disease initiation and progression? Based on their analyses, a general conclusion that SOX4, 11, and 12 are associated with cancer can be made. However, unless a more targeted approach in assessing the SOX genes expression in different organs and stages are made, the usefulness of their results to clinical practice, beyond the SOX genes are associated with cancer, is challenging to evaluate.
  2. Based on their extensive analyses of the SOX family, it will be interesting to hear whether the authors believe that the SOX genes can be considered ubiquitous oncogenes, such as c-myc oncogene, for example. An overwhelming majority of the studies listed in Fig. 1 show SOX4, 11, and 12 are associated with cancer, but some studies do not show that. I think the authors should consider that and should discuss (or at least) mention the reliability of the original studies and the discrepancy between some of these studies.
  3. The authors could have also likely benefitted from a gene network analysis to derive more useful information about these 3 genes. The 3 genes are weakly correlated, and correlations generally do not reflect the strength of potential interaction between genes.

Another point is whether the authors looked at the distribution of the nature of the mutations in SOX4. Are these missense, nonsense, or are they affecting the level of expression of the SOX4 gene? How is the mutational load itself associated with the various cancer phenotypes the SOX4 gene is associated with. 

Round 2

Reviewer 3 Report

I have no additional comments to the authors